NKCC1 involvement in the epithelial-to-mesenchymal transition is a prognostic biomarker in gliomas

Sun Huaiyu
Long Shengrong
Wu Bingbing
Liu Jia
Li Guangyu liguangyu1972@sina.com
Department of Neurosurgery, First Hospital of China Medical University , Shenyang , China
Bandapalli Obul
Electronic publication date: 2020 Mar 16
Publication date: 2020
Volume: 8
Electronic Location ID: e8787
Received 2019 Oct 30; Accepted 2020 Feb 23
Copyright: ©2020 Sun et al.
Copyright year: 2020
Copyright holder: Sun et al.
License: This is an open access article distributed under the terms of the Creative Commons Attribution License, which permits unrestricted use, distribution, reproduction and adaptation in any medium and for any purpose provided that it is properly attributed. For attribution, the original author(s), title, publication source (PeerJ) and either DOI or URL of the article must be cited.
License URL: https://creativecommons.org/licenses/by/4.0/

Keywords: Epithelial-mesenchymal transition (EMT); Glioma; Invasion and migration; Solute Carrier Family 12, Member 2 (SCL12A2/NKCC1)

Funding: Science and Technology project of Shenyang 18-014-4-03 Science and Technology project of Education Department of Liaoning province LFWK201705 The project was supported by Science and Technology project of Shenyang (18-014-4-03) and Science and Technology project of Education Department of Liaoning province (LFWK201705). The funders had no role in study design, data collection and analysis, decision to publish, or preparation of the manuscript.

==============================
Background

Gliomas are the most prevalent type of intracranial tumors. NKCC1 is an important regulator in tumor cell volume. We noticed that abnormally high NKCC1 expression resulted in changes in the shape and adhesion of glioma cells. However, little is known about the role of NKCC1 in the epithelial-mesenchymal transition (EMT) of gliomas. This study aims to clarify the biological function of NKCC1 in glioblastoma multiforme (GBM) progression.

Methods

Using data from The Cancer Genome Atlas (TCGA), we performed a Kaplan–Meier analysis on NKCC1 expression levels to estimate the rate of survival of mesenchymal GBM patients. The correlation between NKCC1 and EMT-related proteins was analyzed from the Gene Expression Profiling Interactive Analysis (GEPIA) server. We conducted Gene Set Enrichment Analysis (GSEA) to verify molecular signatures and pathways. We then studied the expression of NKCC1 in grade I–IV glioma tissue samples collected from patients using immunohistochemistry (IHC). Finally, we evaluated the effects of NKCC1 migration and invasion on the cellular behaviors of U251 cells using the transwell assay and western blots.

Results

High NKCC1 expression was associated with poor prognoses in mesenchymal GBM. Our results suggest a correlation between NKCC1 and EMT-protein markers: CDH2 and VIM. GSEA showed that gliomas, TGF-beta signaling and EMT were enriched in the NKCC1 high expression phenotype. Higher expression levels of NKCC1 in gliomas correlate with higher glioma grades. Transwell assay and western blot results demonstrated that the knockdown of NKCC1 led to a reduction in migration and invasion, while also inhibiting MMP-2 and MMP-9 expression in U251.

Conclusion

These results suggest that high expression of NKCC1 regulates EMT in gliomas, providing a new therapeutic strategy for addressing the spread of gliomas by inhibiting the spread of intracranial tumors.

Introduction

Gliomas are one of the most lethal brain tumors. They affect cognitive function and have a high risk of recurrence, severely affecting quality of life. According to the World Health Organization (WHO), gliomas are classified into grades I–IV. Grades I–II are low-grade gliomas (LGG) and grades III–IV are high-grade gliomas. Among high-grade gliomas, glioblastoma multiforme (GBM) has the highest degree of malignancy and its median survival time is approximately one year (Chen et al., 2017; Louis et al., 2007). Currently, the primary treatment for gliomas is surgical resection followed by radiotherapy, chemotherapy, immunotherapy, photodynamic therapy and electric field therapy (Louis et al., 2016). Despite current treatments, GBM has a high rate of recurrence and a 5-year survival rate of 9.8% (Shen et al., 2015). To improve the prognosis of GBM patients, it is important to identify therapeutic targets that effectively prevent intracranial invasion and tumor proliferation.

We can attribute the high recurrence of GBM to the migration of treatment resistant brain-tumor-initiating cells (Guerrero Cazares et al., 2012; Lathia et al., 2015). During cell migration, GBM cells change their volume to pass through narrow spaces (Watkins & Sontheimer, 2011). These volume changes are mediated by ion co-transporters, including the potassium chloride co-transporter, NKCC1. NKCC1 regulates intracellular volume and Cl− accumulation, allowing Na+, K+ and Cl− to move through the plasma membrane using energy produced by the plasma membrane.

Tumor metastasis is a multi-step process that begins with the tumor cells exiting their primary site to invade healthy cells. Epithelial-mesenchymal transition (EMT) is a crucial process in tumor invasion (Lemieux et al., 2009). EMT is defined by a loss of polarity, tight junctions and adhesion between epithelial cells, resulting in cell infiltration and migration. This process involves changes in the characteristics and morphology of mesenchymal cells (Brandhagen et al., 2013). Epithelial cells play an important role in embryonic development, tissue repair, tumor progress and organ fibrosis. During tumorigenesis, there is a higher rate of EMT in migrating cells (Zhan, Chen & Chen, 2018).

In our study, we hypothesized that high levels of NKCC1 expression in gliomas were positively correlated with the expression of EMT-related markers: CDH2 and VIM. We concluded that a high expression of NKCC1 was related to gliomas, MAPK signaling pathways, TGF-beta signaling, EMT and other pathways and phenotypes. Our experiments showed that NKCC1 tumor expression increased as the tumor grade increased. NKCC1 knockdown reduced the expression of EMT markers. Simultaneously, glioma invasion was significantly weakened in vitro. These results suggest that NKCC1 may serve as a therapeutic target inhibiting the migration and invasion of gliomas.

Material and Methods

Survival analysis of NKCC1 and its correlation with EMT marker gene expression

We used Gene Expression Profiling Interactive Analysis (GEPIA) (http://gepia.cancer-pku.cn/), an interactive website based on The Cancer Genome Atlas (TCGA), and Genotype-Tissue Expression (GTEx) project data for differential expression analysis, correlation analysis, patient survival analysis, similar gene detection and dimension analysis (Tang et al., 2017). Using the GEPIA database, we divided the mesenchymal subtype of GBM into two groups based on the degree of NKCC1 expression and plotted Kaplan–Meier (KM) survival curves. Using GEPIA, a Pearson correlation test was conducted to evaluate the relationship between NKCC1 and EMT proteins in the TCGA-GBM datasets.

Gene set enrichment analysis (GSEA)

GSEA is a computational method used to determine whether a set of previously defined genes show statistically significant and consistent differences between two biological states (Subramanian et al., 2005). We used GSEA to generate an ordered list of genes based on their correlation to NKCC1 expression and to identify survival differences between the high-expression and low-expression of NKCC1. The genes were arranged 100 times per analysis. The expression level of NKCC1 was used as a phenotypic marker. We used the normalized enrichment score (NES) to rank the pathways in each phenotype. Analysis was conducted using default settings. A false discovery rate (FDR) of <0.25 and NOM p-value of <0.1 were considered statistically significant.

Immunohistochemistry

Glioma tissue samples were collected from the First Hospital of China Medical University. This study was approved by the First Hospital of China Medical University ethics committee (IRB No: 2017-98-2). The patients we chose for this study met the following criteria: a. diagnosed with a primary glioma, b. the patient information is complete and contains clinicopathological and prognostic characteristics. All patients provided written informed consent. We used IHC to detect the expression of NKCC1 in paraffin-embedded glioma tissues. The incubation of primary antibodies (No. ab59791 rabbit-anti-NKCC1 WB 1:1000 IHC 1:50) was conducted overnight at 4 °C. The incubation of secondary antibodies was conducted for two hours at room temperature. We used the VECTASTAIN Elite ABC staining system for immune detection. Diaminobezidin (DAB) was used as the substrate for color visualization. Images were obtained using a Nikon TE-2000 Brightfield microscope.

Knockdown of NKCC1

NKCC1 shRNA (shNKCC1 target sequence, 5′-ACACACTTGTCCTGGGATT-3′) and a non-targeting control shRNA sequence were obtained from Takara Biotechnology in Dalian, China. The shRNA sequence was inserted into pRNA-H1.1 to construct pRNA-H1.1-NKCC1 and pRNA-H1.1-control plasmids. GBM cell lines U251/U87 were obtained from the cell bank of the Shanghai Institutes for Biological Sciences, Chinese Academy of Sciences. The Short Tandem Repeat (STR) profiles for these two cell lines are provided in the supplementary files. U251 and U87 cells were transfected with pRNA-H1.1-NKCC1 or pRNA-H1.1-control using Lipofectamine 3000 (Invitrogen). The cells were divided into three groups: a. NC group, control U251/U87 cells; b. Vector group, U251/U87 cells transfected with pRNA-H1.1-control plasmids; c. shNKCC1 group, U251/U87 cells transfected with pRNA-H1.1-NKCC1. These cells were collected for additional experiments 48 h after transfection. The expression of NKCC1 in U251/U87 cells were analyzed using western blot as described below.

Western blot

Cellular protein was extracted using a protein extraction kit, according to the manufacturer’s instructions (Catalog No. WLA019, Wanleibio, China). Proteins were separated using 8%–15% SDS-PAGE gel electrophoresis and transferred to polyvinylidene fluoride (PVDF) membranes. The membranes were blocked with 3%–5% bovine serum albumin (BSA) for an hour at 37 °C. The primary antibodies (N-cadherin rabbit [No. ab18203 WB 1:1000; VIM rabbit] [No. ab45939 WB 1:1000; ZEB1 mouse] [No. ab181451 WB 1:1000; MMP-2 rabbit] [No. ab37150 WB 1:1000; MMP-9 rabbit] [No. ab 38898 WB 1:1000; CTNNB1 rabbit] [No. ab16051 WB 1:1000; β-actin mouse] [No. ab8227 WB 1:500]) were incubated at 4 °C overnight. The following day, PVDF membranes were washed three times using tris-buffered saline tween-20 (TBST). The PVDF membranes were then incubated with a secondary antibody (goat anti-rabbit/mouse IgG 1:2000) for one hour at room temperature and washed three times, using TBST for five minutes each. Finally, Electrochemiluminescence (ECL) and a gel scanner were used to expose bands of PVDF.

Statistical analysis

All data were expressed as mean ± SD for three independent experiments performed in triplicate; p < 0.05 was considered statistically significant. We performed analysis using Statistical Product and Service Solutions (SPSS) 23 and GraphPad Prism 7.0.

Results

The correlation of NKCC1 with mesenchymal GBM and EMT-related proteins with NKCC1 in GBM

Kaplan–Meier survival analysis showed that the prognosis of GBM in the NKCC1-HIGH group was poor compared to the NKCC1-LOW group (log-rank p = 0.068; Fig. 1). Univariate analysis showed that the NKCC1-HIGH expression group was associated with poor overall survival (OS) (hazard ratio [HR]: 1.9). To study the regulatory mechanism of NKCC1 and EMT in GBM, the correlation between the expression of NKCC1 and EMT markers was analyzed using the GEPIA database. As predicted, NKCC1 expression in the TCGA-GBM dataset positively correlated with EMT-related proteins, CDH2, the interstitial marker N-cadherin encoding gene, VIM, Zeb1 and CTNNB1 (beta catenin; Fig. 2). We performed GSEA on low- and high-NKCC1 expression datasets to determine the differentially activated signaling pathway in GBM. There were significant differences in the enrichment of the MSigDB Collection (Hall Mark and KEGG by GSEA). We selected the most significant enriched signaling pathways using NES. We found that gliomas, MAPK signaling pathways, TGF-beta signaling, EMT and other pathways and phenotypes correlated to a high expression of NKCC1. DNA repair, base excision repair, mismatch repair and nucleotide excision repair correlated to a low expression of NKCC1 (Fig. 3).

Figure 1 Impact of NKCC1 expression on overall survival in Mesenchymal GBM patients in TCGA cohort.

Figure 2 Correlations and validations of EMT markers with NKCC1 in GBM.

The Pearson correlations between NKCC1 and the EMT markers, including (A) CDH2, (B) VIM, (C) Zeb1 and (D) CTNNB1 in the TCGA GBM dataset.

Figure 3 Enrichment plots from gene set enrichment analysis (GSEA).

GSEA results showing that (A) Gliomas, MAPK signaling pathways are differentially enriched in NKCC1-related GBM in the GSEA/Hallmark category. (B) TGF-beta signaling and epithelial mesenchymal transitions are differentially enriched in NKCC1-related GBM in the GSEA/KEGG category.

Immunohistochemistry

We used IHC to measure NKCC1 expression in glioma tissues. We found that NKCC1 was highly expressed in high-grade (III and IV) gliomas compared to low-grade (I and II) gliomas (Fig. 4). Table 1 shows the clinicopathological characteristics of patients with gliomas based on their NKCC1 expression status. These findings show that NKCC1 expression levels are related to different glioma grades (p < 0.05).

Figure 4 NKCC1 expression was associated with the histopathological grade in human glioma tissues.

The images of immunohistochemical staining of NKCC1 in human different grade glioma tissues. (A) Grade I glioma tissue. (B) Grade II. (C) Grade III. (D) Grade IV.

Table 1 Clinicopathological characteristics of glioma patients based on NKCC1 expression status.

Characteristic	IHC intensity scoring	p value	
		0	+1	+2	+3		
Sex	Male	4	8	5	10	0.277	
Female	3	7	3	21	
Age	<50	5	6	6	14	0.445	
≥50	2	3	7	17	
Grade	I	1	3	0	1	0.001	
II	4	9	3	3	
III	2	1	4	9	
IV	0	2	1	18	
Location	Left cerebral hemisphere	1	8	4	15	0.62	
Right cerebral hemisphere	6	7	4	15	
Cerebellum	0	0	0	1	

NKCC1 is involved in EMT, invasion and migration of U251 and U87 cells

With a transwell assay we determined the role of NKCC1 in the invasion of U251 and U87 cells using shRNA-transfected U251/U87 cells and control shRNA U251/U87 cells. We counted the number of cell invasions using Matrigel as the reconstituted basement membrane matrix. We cultured cells under conventional conditions and removed them after 24 h. We found that migration was significantly lower in shRNA U251/U87 cells than in the control, regardless of the presence of Matrigel (Fig. 5). U251/U87 cells were cultured at 37 °C and 5% CO2. We constructed short hairpin (SH) RNA and found that it knocked down the expression of NKCC1 in U251/U87 cells. Using western blot analysis to silence NKCC1 generated lower expression of EMT markers, MMP2 and MMP9 in U251/U87 cells, compared to the control group. These results suggest that the invasion and migration of U251/U87 cells was inhibited by NKCC1 (Fig. 6). CTNNB1 protein expression was also significantly decreased in shNkcc1-U87/U251 cells (Fig. 7; * indicates p value <0.05).

Figure 5 NKCC1 promoted the ability of U251 and U87 cells to migrate and invade.

(A–D) Transwell assay showing migration in U87 cell line. (E–H) Transwell assay showing invasion ability in U87 cell line. (I–L) Transwell assay showing migration ability in U251 cell line. (M–P) Transwell assay showing invasion ability in U87 cell line. * indicates P < 0.05.

Figure 6 Silencing of NKCC1 in U87 and U251 as detected by western blotting.

β-actin was used as a positive control. Vim, Zeb1, CDH2, CTNNB1, NKCC1, MMP2 and MMP9 were detected by western blotting after knockdown of NKCC1. (A–B) Representative immunoblots for expression of Vim, Zeb1, CDH2, CTNNB1, NKCC1, MMP2 and MMP9 in U87 and U251 cell. (C–H) Summary data of immunoblotting expression of each protein was first normalized by β-actin and relative expression level in U87 and U251 cell. * indicates P < 0.05.

Figure 7 Silencing of NKCC1 in U87 and U251 as detected by western blotting.

β-actin was used as a positive control. CTNNB1(Catenin beta-1) was detected by western blotting after knockdown of NKCC1. (A) Representative immunoblots for expression of CTNNB1 in U87 and U251 cell. (B) Summary data of immunoblotting expression of CTNNB1 was first normalized by β-actin and relative expression level in U87 and U251 cell. * indicates P < 0.05.

Discussion

Previous studies have reported on the expression and function of NKCC1 in cancer (Sun et al., 2016; Shiozaki et al., 2014; Wright et al., 2009). High NKCC1 expression levels play a role in regulating EMT in gliomas, providing a new therapeutic strategy for addressing the spread of gliomas and inhibiting the spread of intracranial malignancies. The prognostic value of NKCC1 in GBM has not been explored, leading us to focus on the role of NKCC1 in GBM invasion and migration. Using RNA-seq data from TCGA, our bioinformatics analysis showed that high NKCC1 expression in mesenchymal GBM was associated with a lower survival time and a poor prognosis. To further study the role of NKCC1 in GBM, we used the TCGA-GBM dataset for GSEA, and found that a high expression of NKCC1 was associated with gliomas, MAPK signaling pathways, TGF-beta signaling, and EMT. We also found that in GBM, NKCC1 expression positively correlated with EMT marker expression. IHC showed that NKCC1 was highly expressed in gliomas. We found that higher glioma grades correlated with higher NKCC1 expression. Additionally, western blot analysis revealed that the knockdown of NKCC1 expression levels in U251 decreased EMT-related gene expression. Finally, cell invasion assays revealed that the knockdown of NKCC1 significantly reduced glioma invasion and migration. The data suggests that NKCC1 may promote EMT in gliomas. CTNNB1 plays an important role in the Wnt/β-catenin pathway and contributes to EMT in multiple cancers. EMT is regulated by multiple transcription factors including Snai1/2, ZEB1/2, twist1/2 and signaling pathways including TGFβ, ERK1/2, AKT, Notch and WNT/catenin. Therefore, we hypothesize that NKCC1 uses Wnt/β-catenin to promote glioma invasion.

NKCC1 is highly expressed in many forms of cancer, including GBM. The high expression of NKCC1 is often associated with a poor prognosis in GBM patients (Pavón et al., 2015). In our study, the high expression of NKCC1 was observed in both GBM and U251/U87 cells; high mRNA levels of NKCC1 were associated with a poor clinical outcome in mesenchymal GBM patients. The direct involvement of NKCC1 in biological processes has not yet been studied. We confirmed the relationship between NKCC1 and EMT in gliomas using GSEA. The high expression of NKCC1 promoted EMT metastasis by activating the STAT3 signaling pathway in non-small-cell lung cancer (Lin et al., 2017). Although GBM is characterized by local invasion, it rarely produces clinically significant extracranial metastasis and only 0.4% of GBM patients experience metastasis to internal organs. However, there are 20% of GBM patients with detectable levels of tumor cells circulating in their blood  (Awan et al., 2015). Additionally, circulating tumor cells from patients diagnosed with metastatic breast cancer showed different epithelial and mesenchymal phenotypes. Circulatig tumor cells (CTCs) exhibited higher levels of interstitial phenotypes than cancer cells within primary tumors (Yu et al., 2013). Our results suggest that mesenchymal transition is a key molecular event increasing the malignancy of gliomas  (Kahlert, Nikkhah & Maciaczyk, 2013).

NKCC1 is found in the tissues of many different animals and is important in multiple physiological functions, including transporting ions to secrete and absorb epithelial cells, maintaining and regulating cell volume, ion concentration and regulating cell growth and development. The mRNA expression of NKCC1 in the central nervous system is 5 to 40 times higher than in other tissues (Yerby et al., 1997). Emerging evidence suggests that NKCC1 plays an important role in changing cell volume during mitosis. NKCC1 blockers can significantly inhibit the proliferation of glioma cells (Turner & Sontheimer, 2014). NKCC1 is also involved in the process of tumor apoptosis induced by chemotherapeutic drugs and plays an important role in the survival of glioma cells (Algharabil et al., 2012). However, NKCC1’s primary function in tumors is to promote call invasion. Existing evidence suggests that NKCC1 changes the adhesion of cells. The regulation of cell volume requires the movement of water, induced by osmotic pressure gradients, to achieve regulatory volume reduction (regulatory volume decrease, RVD) and regulatory volume increase (RVI). Chloride ions play an important role in this process, and NKCC1 is one of the most important transporters involved  (Zhu et al., 2014). Some studies have shown that the inhibition of WNK kinase and OSR1 kinase upstream of NKCC1 can also reduce intracellular chloride concentration and inhibit the process of RVI in glioma cells, inhibiting cell invasion (Zhu et al., 2014).

We found that a high expression of NKCC1 promotes EMT in GBM. Many studies support our hypothesis (Ma et al., 2019; Haas et al., 2011). We identified a correlation between NKCC1 expression, the TGF-β signaling pathway and MAPK signaling pathways using GSEA. However, we did not determine a definitive and specific regulatory relationship or potential mechanism between NKCC1 and these two pathways when we induced EMT in vivo or vitro experimental conditions. This was one limitation of our study.

Conclusion

NKCC1 promotes migration and invasion of U251/U87 cells. We found that NKCC1 promotes EMT in gliomas. Thus, NKCC1 may act as a potential target for the treatment of malignant gliomas. Other NKCC1 inhibitors that cross the blood–brain barrier may block the NKCC1-promoted EMT process in the brain; these may be used in combination with temozolomide to block the invasion and migration of gliomas.

Supplemental Information

Supplemental Information 1 Raw data for GSEA

Click here for additional data file.

Supplemental Information 2 STR for U87 cell

Click here for additional data file.

Supplemental Information 3 STR for U521

Click here for additional data file.

Supplemental Information 4 Raw gels

Click here for additional data file.

Additional Information and Declarations

Competing Interests

Author Contributions

Human Ethics

Data Availability

The authors declare there are no competing interests.

Huaiyu Sun performed the experiments, authored or reviewed drafts of the paper, and approved the final draft.

Shengrong Long performed the experiments, analyzed the data, authored or reviewed drafts of the paper, and approved the final draft.

Bingbing Wu analyzed the data, prepared figures and/or tables, and approved the final draft.

Jia Liu conceived and designed the experiments, prepared figures and/or tables, and approved the final draft.

Guangyu Li conceived and designed the experiments, authored or reviewed drafts of the paper, and approved the final draft.

The following information was supplied relating to ethical approvals (i.e., approving body and any reference numbers):

This study was approved by the ethics committee of First Hospital of China Medical University (No. 2017-98-2).

The following information was supplied regarding data availability:

The GSEA raw data are available in a Supplemental File.

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
