# Peer review of "NKCC1 involvement in the epithelial-to-mesenchymal transition is a prognostic biomarker in gliomas"

_PeerJ, doi:10.7717/peerj.8787_

## Round 0.1 · original submission · Major Revisions

The authors should show the results check for beta catenin, one of the main candidate in EMT if it plays any role in glioma in addition to carefully addressing all the points raised by the reviewers.

Reviewer 1 ·

Basic reporting

no comment

Experimental design

no comment

Validity of the findings

no comment

Additional comments

Comments to the Author
The manuscript entitled “NKCC1 is involved in the epithelial-to-mesenchymal transition as a prognostic biomarker in glioma” suggests that NKCC1 plays important roles in regulating EMT in gliomas. This is an interesting article which can provide a novel strategy in treating glioma invasiveness. However, there are several concerns should be considered:
Major concerns:
1. The author showed the results in U251 cells. However, there is no any information about the source of U251 cell. Please provide the detailed information.
2. How about the effects of NKCC1 on the crucial factors of EMT, such as E-cadherin, N-cadherin, vimentin, ZEB1, etc?
3. The author should also provide the crucial signal pathway involved in glioma invasiveness to further reveal the mechanism of NKCC1.
Minor concerns:
1. There are some grammar and format mistakes in this manuscript. For example:
Line 41, “plays” should be “played”.
Line 47, there is an extra space behind the word “function”.
Therefore, the author should polish the language of this manuscript by a native English speaker.
2. Please provide the histogram figures for the detection of migration and invasion.

Reviewer 2 ·

Basic reporting

no comment

Experimental design

1. The effect of NKCC1 was only detected in one cell line U251 which made the conclusions not stringent.
2. Migration and invasion effects of NKCC1 was detected by transwell assay and western blot. I suggest author could add RT-qPCR assay for a better supporting of their conclusion.
3. The information of patients should be provided in the ‘Materials and Methods’ section, including gender, ages, stages, etc. The exclusion and inclusion criteria of the patients should be indicated as well.

Validity of the findings

1. Authors should explain why “we did not determine a definite and specific regulatory relationship or potential mechanism between NKCC1 and transforming growth factor beta (TGF-β) induced EMT in vivo or vitro experimental conditions.”. The depth of “Discussion” section is not enough.
2. Authors should discuss is there any target genes involved in the underlying mechanism of NKCC1. The further research should be discussed, otherwise only one pathway will lead to insufficient research depth.
3. P value is statistical result so it should be presented in ‘Result’ part.

---

## Round 0.2 · Minor Revisions

The reviewers found that the manuscript has improved upon revision but still found some issues which needs to be addressed.

Please address them carefully and submit your revision.

Reviewer 2 ·

Basic reporting

no comment

Experimental design

no comment

Validity of the findings

no comment

Additional comments

Comments to the author:
Within this manuscript entitled “NKCC1 is involved in the epithelial-to-mesenchymal transition as a prognostic biomarker in glioma”, authors explored the function and mechanism of NKCC1 in glioma cells. The results shown that high expression of NKCC1 could regulate the EMT in glioma. And NKCC1 overexpression were correlated with glioma grade. This is explored the novel insight of NKCC1. However, there are several minor issues that need clarification and correction.
Materials and Methods
- How long the transfection last?
- How did you quantify the protein concentration in Western blot part? Did you use a kit? In addition, the catalog and dilution of antibodies should be provided.
Results:
-The error bar was missing in Figure7.
- Since author said all data were expressed as mean ± SD, however, in Figures, it was presented as mean + SD.
Discussion:
Author introduced too many experimental results in this part, the depth of discussion should be improved, I suggest author increase the cited references size to combine with previous studies for a in-depth discussion.

---

## Round 0.3 · accepted · Accept

With your implementation of the reviewers and editorial comments the manuscript is suitable for publication in PeerJ.